# Levels of Physical Activity, Family Functioning and Self-Concept in Elementary and High School Education Students: A Structural Equation Model

**DOI:** 10.3390/children10010163

**Published:** 2023-01-14

**Authors:** Félix Zurita-Ortega, José Manuel Alonso-Vargas, Pilar Puertas-Molero, Gabriel González-Valero, José Luis Ubago-Jiménez, Eduardo Melguizo-Ibáñez

**Affiliations:** Department of Didactics of Musical, Plastic and Corporal Expression, University of Granada, 18071 Granada, Spain

**Keywords:** teenagers, physical activity, high school education, elementary education

## Abstract

In the adolescent population there is great concern about low levels of physical activity and low levels of family awareness of the benefits of physical exercise on physical and mental health. This study aims to determine the influence of physical activity levels, family functioning and self-concept in primary and secondary school students, as well as to develop a structural equation model as a function of weekly physical activity time. A descriptive, cross-sectional, comparative study was conducted on whether students engage in more than three hours of physical activity per week. To collect the data, instruments validated by the scientific community were used, such as the Adaptability, Partnership, Growth, Affection, and Resolve family questionnaire (APGAR) and the self-concept questionnaire form 5. The results show that those students who engage in more than 3 h of physical activity per week have higher levels of family functioning than those who do not meet this sport criterion. In addition, physically active students show higher scores on all dimensions of self-concept than those who practice less than 3 h of physical activity per week. Finally, as a conclusion, it can be affirmed that the amount of physical activity practice brings benefits to student’s mental health.

## 1. Introduction

Adolescent lifestyles have been increasingly sedentary for decades [1,2,3,4]. This is in contrast to the evidence of the benefits of physical activity for people in areas such as their social life, by promoting autonomy and integration [5,6,7], their physiological and psychological wellbeing, by improving physical and mental well-being [8,9] and, in particular, their self-esteem [10]. Despite this reality, young people show a decrease in their interest in physical activity as their development approaches the adolescent stage [11], in favour of more sedentary activities, such as the use of video games and other electronic devices [10]. This makes this population group one of the least active [12].

This physical inactivity has led to an increase in cardiovascular diseases, increasing obesity levels worldwide in the adolescent population [4]. Physical inactivity has also been shown to be detrimental to the development of psychological and psychosocial factors [1]. These unhealthy habits, together with a lower intake of physical activity, can psychologically affect aspects such as the individual’s self-concept [13]. In adolescents, numerous studies show a positive relationship between physical activity levels, self-esteem and self-concept [14,15]. 

Self-concept is defined as a person’s appreciation of his or her physical appearance [16]. In this case, this variable is positively influenced by physical activity [9]. This can lead to lower levels of anxiety because people feel more satisfied with their physique [17,18]. Similarly, social relationships, especially family relationships, contribute favourably to the student’s self-concept [16,19]. In this sense, self-concept has a direct impact on the identity and behavioural aspect of people [20] as well as on academic performance [21]. Currently, self-concept is understood as a multidimensional domain that harbours specific aspects within it [22,23,24]; following the model of Shavelson et al. [23], its internal structure is composed of academic self-concept and non-academic self-concept, the latter including physical, social and emotional self-concepts more specifically [25]. Based on this multidimensional self-concept model, García and Musitu [26] determined five specific dimensions: academic, social, emotional, physical and family. All these dimensions interact with each other, influencing the person’s overall self-image [23,26].

The family environment also contributes significantly to children’s adherence to appropriate lifestyle habits [27,28], to adequate mental health [29] and to the transmission of values, norms, knowledge and behavioural patterns [30]. This is because family members are interconnected and this composition possibly influences them [30,31,32]. Another reason why the nuclear family is of vital importance is because it aids in exploring some of the different human behaviours [33,34]; in children, this influence is reproduced as an educational and behavioural model [35,36]. Low levels of healthy habits and negative family behaviours can lead to negative social behaviour, poor school performance, inadequate emotional development and even promote the development of nutrition-related diseases [37,38,39,40]. Similarly, family deterioration in childhood can lead to negative well-being in the future [29]. Along these lines, better family functioning contributes to a higher level of moderate and vigorous physical activity [41].

Taking into account the possible interaction between the study variables, the aim here is to determine the influence of physical activity levels, family functioning and self-concept in primary and secondary school students, as well as to develop a structural equation model regarding physical activity time per week.

The following research hypotheses are proposed:

**H.1.** 
*Students who practice more than 3 h of weekly physical activity will obtain higher scores in family functioning and in the different areas of self-concept.*


**H.2.** 
*Adolescents who practice more than 3 h of physical activity per week will show a better effect of family functioning on the different areas of self-concept.*


## 2. Materials and Methods

### 2.1. Design and Participants

A descriptive, comparative, cross-sectional and non-experimental (ex post facto) research study was carried out with students from the province of Granada, Spain. In this case, the schools were randomly selected. The sample size consisted of 706 participants. The average age of the participants ranged from 11 to 15 years old (Age = 12.01; SD = 1.13). The gender distribution was 56.1% (n = 396) male and 43.9% (n = 310) female. In this case, the sampling error study, for a confidence level of 95%, yielded an error of less than 5.0%.

### 2.2. Instruments

**Sociodemographic questionnaire:** This instrument was designed to collect sociodemographic and physical sporting variables. The socio-demographic variables were sex (male/female) and age. To collect the physical-activity time, the following question was used: do you do more than 3 h of physical activity outside school hours? This question has already been used by other studies to measure whether young people are physically active outside school hours [42]. This variable has an ordinal character, as its values represent categories with some intrinsic ranking.

**APGAR questionnaire:** This instrument has been used to measure the degree of family functioning. This questionnaire has been developed by Austin and Huberty [43]; however, for this research we have used the version adapted to Spanish and to the study population by Suárez and Alcalá [44]. This is made up of a total of five questions, including "do you feel that your family loves you?”, through which the degree of family functionality is measured. Regarding the reliability analysis of the instrument, Cronbach’s Alpha showed a value of α = 0.790. This variable has an ordinal character.

**Self-Concept Questionnaire Form 5:** This questionnaire was developed by García and Musitu [45] and this version was used to collect the data for this research. This questionnaire assesses self-concept from a multidimensional perspective, dividing self-concept into five dimensions: academic, emotional, social, physical and family. Regarding the degree of reliability, the instrument showed a value of α =0.839. This variable has a scaled character, as its values represent ordered categories with a meaningful metric.

### 2.3. Procedure

Prior to carrying out this research, a systematic search was carried out on the subject to study the most reliable instruments for data collection, as well as to guide this research. Then, once the sample group had been selected, the different educational institutions were contacted and invited to collaborate in this research. Once a favourable response was obtained from the educational establishments, the research team drafted an informative letter from the department of Didactics of Musical, Plastic and Bodily Expression, which was addressed to the legal guardians of the children, authorising their sons and daughters to participate in the study. The researchers were present during data collection, helping to resolve any doubts that arose during the completion of the questionnaires.

To avoid possible random responses, one question was duplicated, eliminating all participants where this question did not match. In this case, a total of 12 responses were eliminated. In terms of ethics, the study followed the principles set out in the Declaration of Helsinki. The study was supervised at all times by an ethics committee at the University of Granada (2966/CEIH/2022).

### 2.4. Data Analysis

To carry out a comparative analysis of the results, the IBM SPSS Statics programme in version 25.0 (IBM Corp, Armonk, NY, USA) was used. Prior to the comparative study, the normality of the sample was studied using the Kolmogorov–Smirnov test, and a normal distribution was found.

In this case, the comparative analysis was carried out using the T-Student test for independent samples. For the study of statistically significant differences, the Pearson Chi-square test was used, establishing the level of significance at *p* ≤ 0.05. For the effect size, Cohen’s standardised *d* measure [46] was used. It is interpreted as follows: null (≤0.19), small (0.20–0.49), medium (0.50–0.79) and large (≥0.80). The Levene test was carried out in order to evaluate the equality of the variances for a variable that presents two or more groups. After checking that the family functionality variable followed a non-normal distribution, the data were analysed using the Mann–Whitney U test.

IBM SPSS Amos version 26.0 software (IBM Corp, Armonk, NY, USA) was used to develop the structural equation models. The proposed models allow the correlation of the variables to be studied in terms of whether the participants meet the established physical activity criteria. Observing the model (Figure 1), it is composed of unidirectional arrows in such a way that the effect of family functionality on the different areas of self-concept is studied. Focusing on the type of variables, the model is composed of a single exogenous variable (FF) and five endogenous variables (AC-SC; SO-SC; FA-SC; PH-SC; EM-SC). In order to include the measurement of observed variables, causal relationships have been examined on the basis of the degree of reliability of the measurements and on the basis of the indicators. For the level of significance, two levels have been established: the first for *p* ≤ 0.05 and the second for *p* ≤ 0.001.

To evaluate the different equation models, the criteria established by Bentler [47] have been followed. Each of the models must be adjusted in terms of the fit indices, each of these evidencing a value greater than 0.900. Another value to take into account is the root mean square error of approximation (RMSEA), which must show a value lower than 0.100. Finally, an isolated interpretation of the results obtained cannot be performed due to the degree of statistical sensitivity, so other adjustment indices have been used [48].

## 3. Results

Table 1 shows the comparative study of those young people who practice more than 3 h of physical activity per week outside school hours and those who do not meet this physical sports criterion. For the latter group, higher academic self-concept scores are observed (3.39 ± 0.80). In contrast, participants who do meet this physical sport criterion show higher scores in family functioning (2.80 ± 0.50), social self-concept (4.10 ± 0.64), family self-concept (4.40 ± 0.73), physical self-concept (3.65 ± 0.82) and emotional self-concept (3.45 ± 0.72).

Table 2 shows the comparative study between the degree of family functionality and the weekly physical activity time. Observe how the participants who practice more than 3 h of weekly physical activity show a higher level of family functionality (M = 2.805) than those who practice less than 3 h (M = 2.715).

The proposed structural equation model for participants who engage in more than three hours of physical activity per week has shown a good fit for each of the fit indices (Figure 2 and Table 3). In this case, the Chi-square adjustment obtained a non-significant value (X^2^ =5.648; df = 16; pl = 0.001). In terms of the different fit indices, values of 0.990, 0.910 and 0.901 were obtained for the comparative fit index (CFI), the normalised fit index (NFI) and the incremental fit index (IFI). For the Tucker–Lewis index (TLI), a value of 0.955 was obtained; finally, for the root mean square error of approximation (RMSEA) a value of 0.087 was obtained.

The above figure and table show the results obtained for participants who practice more than 3 h of physical activity per week. In this case, a positive correlation of family functionality on academic self-concept (*p* ≤ 0.001; r = 0.210), social self-concept (*p* ≤ 0.05; r = 0.165), family self-concept (*p* ≤ 0.001; r = 0.715), physical self-concept (*p* ≤ 0.001; r = 0.326) and emotional self-concept (r = 0.022) is observed.

The proposed structural equation model for participants who engage in more than three hours of physical activity per week has shown a good fit for each of the fit indices (Table 4 and Figure 3). The Chi-square adjustment obtained a non-significant value (X^2^ =5.792; df = 16; pl = 0.001). Looking at the different fit indices, for the comparative fit index (CFI), the normalised fit index (NFI) and the incremental fit index (IFI), values of 0.969, 0.902 and 0.900 were obtained. Likewise, for the Tucker–Lewis index (TLI), a value of 0.945 was obtained and finally for the root mean square error of approximation (RMSEA), a value of 0.093 was obtained.

The figure and table above evidence the results obtained for participants who practice less than 3 h of physical activity per week. In this case, a positive correlation of family functionality on academic self-concept (*p* ≤ 0.001; r = 0.301), social self-concept (*p* ≤ 0.05; r = 0.106), family self-concept (*p* ≤ 0.001; r = 0.722), physical self-concept (*p* ≤ 0.001; r = 0.254) and emotional self-concept (*p* ≤ 0.05; r = 0.118) is observed. In this case, the structural equation models show that participants who practice more than 3 h of physical activity per week show a greater correlation of family functioning on social self-concept and physical self-concept. On the contrary, it is shown that participants who do not fulfil this physical sports criterion show a greater correlation of family functioning on academic self-concept, family self-concept and emotional self-concept.

## 4. Discussion

This research shows the existing links between family functionality and self-concept in terms of the levels of physical activity practice of elementary and higher education students. It is observed that the results obtained correspond to the objectives set out, so this discussion continues with the intention of showing a comparison between these results and others obtained in previous studies.

In reference to self-concept levels, students with more than three hours of physical activity per week obtain better scores in the dimensions of social, family, physical and emotional self-concept, which corresponds to the results obtained by Ferrari et al. [49] and González-Valero et al. [50]. For the academic self-concept dimension, the highest scores are linked to participants who practice less than three hours of physical activity per week, which does not correspond to the studies by Carriedo et al. [51] and Gedda et al. [52]. The studies cited above [51,52] conclude that there is no relationship between physical activity and academic self-concept. This could be due to the fact that this group gives more importance and time to academic performance than to other activities such as physical activities, with participants who are more physically active being more prone to social and family relationships and to perceive themselves as physically better.

On the other hand, the relationship between physical activity and family functionality shows a positive relationship, assimilating the findings of Kitzman-Ulrich et al. [53] and Lebron et al. [54] who affirm a positive relational tendency of these aspects in their studies, so that people who are physically active are more comfortable in their relationships with their families.

In terms of family functioning, a positive correlation on the dimensions of self-concept was found in both participants who practiced more than three hours of physical activity per week and those who did not. These data correspond to those shown in the studies by Izzo et al. [55], indicating that the feeling of satisfaction with family relationships generates a good self-perception thanks to the support received by those close to them.

Participants with more than three hours of physical activity per week showed a higher correlation of family functioning on social self-concept, coinciding with previous studies that related physical activity levels with family functioning [54,56], physical activity levels with social self-concept [50,57] and family functioning with social self-concept [58,59]. These data show that the practice of physical activity favours familial and social relationships, which may be due to a better feeling of well-being and security in the individual that is extrapolated to his or her environment.

In line with the above, the group that practices more physical activity showed greater significance between family functionality and physical self-concept, as shown in the study by Padial-Ruz et al. [56], thus demonstrating a greater perception of their physique in students with functional families, which could be derived from the practice of leisure activities in the family with contents related to sport or body mobility. In contrast, participants who did not complete three hours of physical activity per week showed a greater correlation of family functionality on academic and family self-concept, which is similar to that reported by Povedano-Diaz et al. [59], as well as with emotional self-concept, as indicated by previous studies in this field [60]. This relationship may be due to the fact that students prioritise and devote more effort to maintaining solid family relationships, good academic performance and emotional stability to the detriment of regular physical activity.

In view of these findings, the educational sphere plays a fundamental role. The study by González-Valero et al. [61] establishes that physical education teachers should encourage healthy behaviours in the youth and adolescent population. Likewise, in the educational environment, the family nucleus plays a key role in encouraging a healthy lifestyle [62]. It has been shown that family motivation towards physical-sports practice together with the family’s socio-economic level act in favour of a healthy lifestyle [63]. The intervention programme carried out by Nyberg et al. [63] found that the family environment helps to increase levels of physical sporting activity, with marked improvements in various psychosocial and physical aspects.

Finally, a new branch of study that is being developed and which can be derived from this research is the pressure of the media on the acquisition of active and healthy lifestyles. In this case, research by Melguizo-Ibáñez et al. [64] and González-Valero et al. [65]] affirm that the media can be used as an element that favours a healthy lifestyle, a statement that is reaffirmed by González-Vallés et al. [66].

## 5. Limitations and Future Perspectives

The research has clearly responded to the proposed objectives; however, there are a series of limitations that need to be pointed out.

The first limitation lies in the typology of the study, since following a cross-sectional design only allows us to study the effect of the data at the time they were collected. It should also be noted that the instruments used, despite having been validated by the scientific community, show an intrinsic measurement error. Furthermore, generalisations cannot be made to a wider national area, as the students belong to a very specific geographical area.

In carrying out this study, a number of future branches of study have been opened up. One of them lies in carrying out a longitudinal study to study how the incidence of the degree of family functionality acts on physical-health and psychosocial variables.

## 6. Conclusions

This research highlights the benefits of practicing more than three hours of physical activity per week during adolescence, as it is observed that participants who meet this physical sports criterion have higher family functionality scores. There are also higher scores in the social, family, physical and emotional areas of self-concept. With respect to the structural equation models, a greater correlation of family functionality on the different dimensions of self-concept is observed in those adolescents who do not practice more than three hours of physical activity per week.

Therefore, due to the benefits observed, the benefits of an active lifestyle on the physical-healthy sphere should be worked on together with families in the educational sphere to help improve the mental health of young adolescents.

## Figures and Tables

**Figure 1 children-10-00163-f001:**
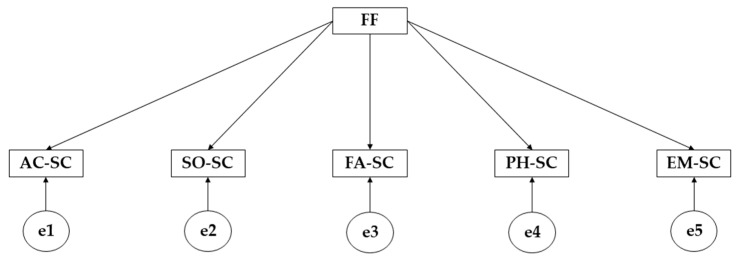
Proposed structural equation model. **Note:** Family functionality (FF); Academic self-concept (AC-SC); Social self-concept (SO-SC); Family self-concept (FA-SC); Physical self-concept (PH-SC); Emotional self-concept (EM-SC).

**Figure 2 children-10-00163-f002:**
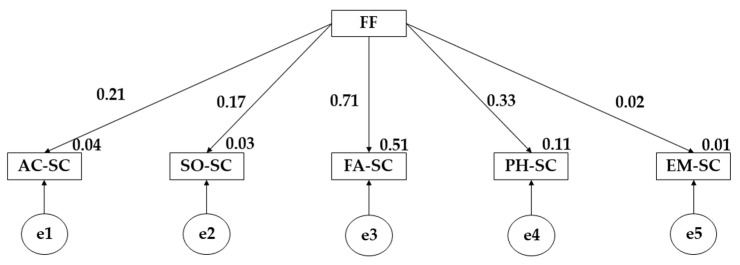
Model developed for students who practice more than 3 h of PA per week. Note: Family functionality (FF); Academic self-concept (AC-SC); Social self-concept (SO-SC); Family self-concept (FA-SC); Physical self-concept (PH-SC); Emotional self-concept (EM-SC).

**Figure 3 children-10-00163-f003:**
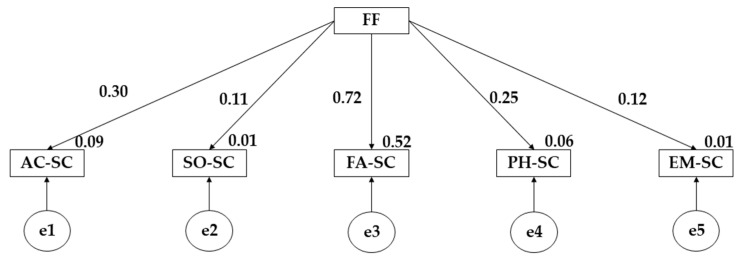
Model developed for students who practice less than 3 h of PA per week. **Note:** Family functionality (FF); Academic self-concept (AC-SC); Social self-concept (SO-SC); Family self-concept (FA-SC); Physical self-concept (PH-SC); Emotional self-concept (EM-SC).

**Table 1 children-10-00163-t001:** Comparative study of the sample according to the physical activity time per week.

		Levene Test					*t*-Test			ES	95% CI
		N	M	SD	F	Sig	T	Gl	P		
**AC**	More than 3 h	340	3.25	0.86	1.15	>0.05	−2.28	688.53	≤0.05	0.16	[0.02; 0.31]
	Less than 3 h	366	3.39	0.80							
**SO**	More than 3 h	340	4.10	0.64	5.85	≤0.05	3.88	703.71	≤0.05	0.295	[0.14; 0.44]
	Less than 3 h	366	3.90	0.71							
**FA**	More than 3 h	340	4.40	0.73	13.43	≤0.05	2.48	700.93	≤0.05	0.190	[0.04; 0.33]
	Less than 3 h	366	4.25	0.84							
**PH**	More than 3 h	340	3.65	0.82	0.19	>0.05	9.38	698.14	≤0.05	0.856	[0.70; 1.01]
	Less than 3 h	366	3.07	0.51							
**EM**	More than 3 h	340	3.45	0.72	1.66	>0.05	4.29	703.99	≤0.05	0.333	[0.18; 0.48]
	Less than 3 h	366	3.20	0.78							

**Note:** Academic self-concept (AC-SC); Social self-concept (SO-SC); Family self-concept (FA-SC); Physical self-concept (PH-SC); Emotional self-concept (EM-SC).

**Table 2 children-10-00163-t002:** Comparative study between time spent in physical activity and family functioning.

		Levene Test			
		N	M	SD	Sig
**FF**	More than 3 h	340	2.80	0.50	≤0.05
	Less than 3 h	366	2.71	0.57	

**Note:** Family functionality (FF).

**Table 3 children-10-00163-t003:** Results of the proposed structural model for participants practicing more than 3 h of PA per week.

Associations between Variables	RW				SRW
	Estimates	SE	CR	*p*	Estimates
AC-SC←FF	0.363	0.092	3.963	***	0.210
SO-SC←FF	0.214	0.069	3.088	**	0.165
FA-SC←FF	1.044	0.056	18.805	***	0.715
PH-SC←FF	0.539	0.085	6.341	***	0.326
EM-SC←FF	0.032	0.079	0.431	0.680	0.022

**Note 1:** Standardised regression weights (SRW); Regression weights (RW); Estimation error (SE); Critical ratio (CR). **Note 3:** *** *p* ≤ 0.001; ***p* ≤ 0.05.

**Table 4 children-10-00163-t004:** Results of the proposed structural model for participants practicing less than 3 h of PA per week.

Associations between Variables	RW				SRW
	Estimates	SE	CR	*p*	Estimates
AC-SC←FF	0.417	0.069	6.039	***	0.301
SO-SC←FF	0.130	0.064	2.033	**	0.106
FA-SC←FF	1.051	0.053	19.959	***	0.722
PH-SC←FF	0.357	0.071	5.008	***	0.254
EM-SC←FF	0.159	0.070	2.275	**	0.118

**Note 1:** Standardised regression weights (SRW); Regression weights (RW); Estimation error (SE); Critical ratio (CR). **Note 3:** *** *p* ≤ 0.001; ***p* ≤ 0.05.

## Data Availability

The data used to support the findings of the current study are available from the corresponding author upon request.

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
