# Peer review of "Levels of Physical Activity, Family Functioning and Self-Concept in Elementary and High School Education Students: A Structural Equation Model"

_children, 2023, doi:10.3390/children10010163_

Round 1

Reviewer 1 Report

METHODOLOGY:

Instruments: It would be pertinent for the authors to deepen the description of each instrument, trying to detail the type of variable generated by its application (quantitative, nominal or ordinal), in order to facilitate understanding by readers less identified with these instruments.

Data Analysis: It is mentioned and that, preceding the comparative analyses, the Kolgomorov-Smirnov test was run, which attested to the normality in the data distribution, but this observed normality is relative to which of the variables under study? Assuming that only the self-concept variable is of a quantitative nature, the normality of the distribution found refers to this.

RESULTS

Table 1: (page 4, line 153) It would be advisable refer  the equation of the test carried out (T-Test), regarding to the Academic Self-Concept dimension, mainly because it represents a tendency contrary to that observed in the relationship between the level of Physical Activity frequency practice and the other Self-Concept variables/dimensions evaluated.

(page 4, lines 154-156) Why the effect size issue was not mentioned from the data presented in Table 1, regarding the Physical Self-Concept, once there was found a strong effect size of phsycal activity frequency level over this variable/dimension? Based on the calculated effect size (Sawilowsky, 2009), there is a probability of 72.8% that a person, drawn at random from the group that practices 3 or more hours of physical activity per week, has a superior physical self-concept that of a person with a lower practice frequency.

(page 5, lines 184-186) Pearson's test translates how the variances of two quantitative variables can be directly or inversely correlated without necessarily pointing to any predictive effect of one over the other, so the term "positive effect" should be reformulated, as it does not represent, actually, an effect relation. Furthermore, for the correlation test, a sub-sample was used - in this case, only those participants who fulfilled the regular physical activity criteria (3 or more hours per week) -, the normality of the distribution of this sub-sample and the existence of linearity in the relationship between Family Functionality and the other dimensions of self-concept were both tested? If so, shouldn´t the fulfillment of these assumptions be described in the methodology.

(page 5, line 190) Since the value of the Chi-Square test was not significant, I ask whether the authors consider relevant to present it, given this value translates a probability of the observed differences could be attributed to chance.

(page 5, lines 209-2011) Same comment regarding page 5, lines 184-186.

DISCUSSION

(page 6, lines 219-220) Given the fact that, in the section dedicated to the description of the methodology, it is mentioned that the average age of the sample under study is 12.01 years old, with a standard deviation of +/- 1.13, it seems strange to me that there is reference to students from higher levels of education.

(page 6, lines 224-233) The authors should be cautious when discussing the results obtained, regarding a possible relationship between self-concept and the regular practice of physical activities, as the literature suggests that it is not clear whether participation in physical activities leads to the development of self-concept, or, on the other hand, people with higher levels of self-concept tend, for this reason, to consider the regular practice of physical activities an important aspect for their quality of life. The strength of the relationship between physical activity and self-concept (and its dimensions) depends on how the data are treated, that is, on how the dependent and independent variables are defined, existing, however, conflicting evidence in the literature on associations of this nature.

(page 7, lines 242-243) Two issues that may bias the parallelism made with the Rohany et al (2011) study, namely the fact Family Functionality was not evaluated with the same instrument, raising here issues of sensitivity of the instruments used to evaluate the construct, and the fact that the samples have peculiarities that demand caution, since the authors are comparing students that have a normal social life with young institutionalized delinquents and that, therefore, may have different conceptions about their families functioning. 

Reviewer 2 Report

There is 38% plagiarism in the manuscript, kindly improved

It seems that translation has been occurred for the whole manuscript.

Most of the sentences are paraphrased meanings are not clear.

Abstract

Rephrase the line 11 & 12. (first sentence).

Line 12 remove o (aims to o determine).

Line 15 & 16 study was carried out on the basis of……? (

Line 17 APGAR used first time – provide full form

Line 18 young people who practice – change to the students.

Line 19 higher can be used while compare between two, other one is miss out for the sentence.

Line 20 remove by way of conclusion

Introduction

Line 29 & 30 What do you mean by young people approach adolescence ?

Materials and methods

Design and participants:

Line 75, the mean and SD (M = 12.01; SD = 1.13) are calculate from where and for what purposes. Justify?  

The physical-sport variables- Not a clear term

This instrument has been used to measure the degree of family functioning. Likewise (should provide the example)

educational centres or educational institutions?

Why were SPSS two versions (25 & 26) used?

sport physique (body) criteria?

physical-sports criteria?

physical activity criteria?

Levene Test was used for what purposes? not mentioned.

Decimal units should be similar

Table 1 did not mention which physical-sports criteria established?

G1 stand for what?

Line 190 The Chi-Square adjustment obtained a non-significant value. (X2 =5.792; df = 16; pl = 0.001). sentence define non-significant value whereas p=0.001 or less than 0.05 (this is the contradict statement or p evidence).

Line 213 & 214 stated greater effect of family functioning on social self-concept and physical self-concept, whereas social self-concept (r=0.106) and physical self-concept (r=0.254) have poor values than others family self-concept (r=0.722), academic self-concept (r=0.301)?

Authors claimed that higher scores in the social, family, physical and emotional areas of self-concept. Higher from whom not discuss, not provide evidences

Round 2

Reviewer 1 Report

Authors' Response1: Thanks for your reply. However, allow me to disagree with it, since, as described in your manuscript, the assessment of Family Functioning is carried out using an instrument (APGAR) in which the assessed person quantifies the degree of frequency certain feelings are raised on him/her by the way her family behaves towards him/her in certain situations, which implies, in my opinion, that it is an ordinal variable and that, therefore, by its nature, does not present normal distribution and so it should be statistically analyzed through non-parametric tests. Hence my question.

Authors' Response 2: Thanks for your reply. I understand your choice, but since the effect size was calculated through Cohen's D and, in the case of students who did not meet the weekly physical activity criterion, an effect was observed on their academic self-concept, it would be interesting, even for your discussion, that this data be highlighted.

Authors' Response 3: Thanks for your reply. However, let me then suggest that the study by Ruiz et al. (2010) be cited in your discussion, as a way to justify your choice of using the term "effect" instead of "correlation".

Authors' Response 4: Thanks for your reply. In this case, then, allow me to suggest that it be briefly explained in a footnote, for example, how the Spanish education system works, taking into account that, in some countries, the expression "secondary education" refers to students in the age group from 15-18 years old. Regarding the 20 students who attended higher levels of scholling, was their exclusion considered in the study, considering that this could bias the results, due to potential differences in maturity that could influence their conceptions about the functioning of their families?
